# Our Experience with SARS-CoV-2 Infection and Acute Kidney Injury: Results from a Single-Center Retrospective Observational Study

**DOI:** 10.3390/healthcare11172402

**Published:** 2023-08-26

**Authors:** Victoria Birlutiu, Bogdan Neamtu, Rares-Mircea Birlutiu, Andreea Magdalena Ghibu, Elena Simona Dobritoiu

**Affiliations:** 1Faculty of Medicine, Lucian Blaga University of Sibiu, Romania, Str. Lucian Blaga, Nr. 2A, 550169 Sibiu, Romania; 2County Clinical Emergency Hospital, Bvd Corneliu Coposu, Nr. 2-4, 550245 Sibiu, Romania; 3Pediatric Research Department, Pediatric Clinical Hospital Sibiu, Str. Pompeiu Onofreiu, Nr. 2-4, 550166 Sibiu, Romania; 4Clinical Hospital of Orthopedics, Traumatology, and Osteoarticular TB Bucharest, B-dul Ferdinand 35–37, Sector 2, 021382 Bucharest, Romania

**Keywords:** SARS-CoV-2 infection, COVID-19, AKI, acute kidney injury, outcome, risk factors

## Abstract

Background: Renal failure in COVID-19 patients is reportedly related to multiple factors such as a direct SARS-CoV-2 cytopathic effect, cytokine storm, the association of pulmonary and/or cardiovascular lesions, the presence of thrombotic microangiopathy, endothelial damage, or the use of potentially nephrotoxic medications. Methods: We retrospectively analyzed 466 cases of SARS-CoV-2 infection, comparing 233 patients with acute kidney injury (AKI) with 233 patients without AKI in terms of their demographic characteristics, comorbidities, clinical background, laboratory investigations, time of AKI onset, therapy, and outcomes after using univariate analysis and a CART decision-tree approach. The latter was constructed in a reverse manner, starting from the top with the root and branching out until the splitting ceased, interconnecting all the predictors to predict the overall outcome (AKI vs. non-AKI). Results: There was a statistically significant difference between the clinical form distribution in the two groups, with fewer mild (2 vs. 5) and moderate (54 vs. 133) cases in the AKI group than in the non-AKI group and more severe and critical patients in the AKI cohort (116 vs. 92 and 60 vs. 3). There were four deaths (1.71%) in the non-AKI group and 120 deaths in the AKI group (51.5%) (*p*-value < 0.001). We noted statistically significant differences between the two study groups in relation to different tissue lesions (LDH), particularly at the pulmonary (CT severity score), hepatic (AST, ALT), and muscular levels (Creatine kinase). In addition, an exacerbated procoagulant and inflammatory profile in the study group was observed. The CART algorithm approach yielded decision paths that helped sort the risk of AKI progression into three categories: the low-risk category (0–40%), the medium-risk category (40–80%), and the high-risk category (>80%). It recognized specific inflammatory and renal biomarker profiles with particular cut-off points for procalcitonin, ferritin, LDH, creatinine, initial urea, and creatinine levels as important predictive factors of AKI outcomes (93.3% overall performance). Conclusions: Our study revealed the association between particular risk factors and AKI progression in COVID-19 patients. Diabetes, dyspnea on admission, the need for supplemental oxygen, and admission to the intensive care unit all had a crucial role in producing unfavorable outcomes, with a death rate of more than 50%. Necessary imaging studies (CT scan severity score) and changes in specific biomarker levels (ferritin and C-reactive protein levels) were also noted. These factors should be further investigated in conjunction with the pathophysiological mechanisms of AKI progression in COVID-19 patients.

## 1. Introduction

Renal impairment in COVID-19, far from being fully assessed, may occur at different time points from the star of the infection. Some patients present with acute kidney injury (AKI) upon admission. Hence, elevated levels of blood urea nitrogen (BUN) (14.4%) and creatinine (13.1%) are recorded from the very beginning [1]. Other patients develop BUN retention during the disease course, with renal impairment worsening as respiratory distress increases. Approximately 20–50% of patients monitored in intensive care units (ICUs) generally develop some form of renal impairment during hospitalization. In these cases, the death rate is approximately 50% [2]. According to a recent meta-analysis by Sabaghian T et al. 2022, the incidence of AKI was 8.9% [3]. The prevalence of acute kidney injury associated with COVID-19 is estimated to be between 0.5 and 85% [4], with a mortality rate of approximately 93% [5].

From a pathophysiological perspective, the complex mechanisms triggering renal failure are reportedly related to multiple factors such as a direct SARS-CoV-2 cytopathic effect, cytokine storm, the association of pulmonary and/or cardiovascular lesions, the presence of thrombotic microangiopathy, endothelial damage, or the use of potentially nephrotoxic medication. Nevertheless, when it comes to the key players involved in renal injury, significant roles have been attributed to proinflammatory status early in the evolution of infection, late-occurring procoagulant status, or cardiac toxicity, which might ensue in the disease’s course [6].

Even though recent meta-analyses have systematically covered the risk factors associated with AKI, such as cardiovascular diseases (hypertension, acute coronary syndrome, shock), respiratory diseases, hepatic insufficiency, diabetes, hyperlipidemia, chronic renal diseases, the reported significance and relevance of each associated factor is different [3,7]. Moreover, many studies in the literature have referred to case reports regarding a limited number of patients with different characteristics [3]. Furthermore, a detailed analysis of the poorly understood pathophysiological mechanisms underlying AKI progression in COVID-19 patients remains difficult to find in the existing literature. As a result of this, in carrying out this retrospective observational study, we aimed to assess the risk factors associated with this outcome in AKI patients and to compare them with the recorded values in non-AKI patients. We hypothesized different patterns regarding demographics, clinical data, and biological data for the two groups and explored the possibility of linking the essential predictors within these patterns based on the decision rules for each group using a machine-learning approach. Several retrospective study designs for forecasting AKI outcomes in COVID-19 patients have been proposed; however, reports making use of CART decision trees are scarce in the literature. To achieve our aim, we propose a 10-fold cross-validation model to assist in disease progression assessment [8,9,10]. 

## 2. Materials and Methods

### 2.1. Data Collection

We selected our cohort based on the single-center electronic records of the Sibiu Clinical County Hospital; the records contained data from between April 2020 and November 2021, during waves 1–4 of the COVID-19 pandemic. 

The County Clinical Emergency Hospital Sibiu, Romania, is a multidisciplinary academic hospital with 1054 beds. It is one of many national hospitals that treated COVID-19 patients during the pandemic, even from the start of the COVID-19 pandemic. The hospital also treated more complex cases from the surrounding counties. The maximum capacity for the management of COVID-19 patients was approximately 160 beds, and 28 ICU beds were dedicated to COVID-19 patients, in addition to the 24 non-COVID-19 ICU beds and beds dedicated to other departments (cardiology, neurology, obstetrics). 

All of the patients admitted in the aforementioned period with positive COVID-19 tests and acute renal failure were selected (RNA positive for SARS-CoV-2) [8,9,10]. Medical records of these patients, which contained details such as their demographic characteristics, risk factors/comorbidities, laboratory investigations, treatments, outcomes in terms of survivability, and complications, were retained. Two independent reviewers revised data retrieval and consistency. Our study was conducted in accordance with the principles of the Declaration of Helsinki and was approved by the Institutional Ethics Committee number (22549/18 September 2020).

### 2.2. Study Population

From the available cohort of 3204 health records, we selected 466 cases, balanced by gender and age. We aimed to retrospectively evaluate a group of 233 patients with AKI, with biochemically elevated urea and creatinine, versus 233 patients without AKI, analyzing them in terms of demographics, comorbidities, clinical background, laboratory investigations, time of AKI onset in the first or second/third week, and therapy-antiviral, immunomodulatory, antibiotic, antifungal, etc., the evolution of the cases. 

Eligible patients met cumulatively the following criteria: (i) Adult patients aged over 18 years and (ii) patients diagnosed with SARS-CoV-2 infection via real-time reverse transcription-polymerase chain reaction (RT-PCR) positive for SARS-CoV-2, and (iii) patients with AKI during the study period (April 2020 and November 2021). Exclusion criteria: (i) patients with chronic kidney disease and/or (ii) patients known to have previous renal replacement therapy (prior to admittance) and/or (iii) patients whose records lacked proinflammatory markers and/or (iv) patients with asymptomatic forms of COVID-19, and/or (v) all other cases that could not be balanced by age and gender with the AKI group. The inclusion and exclusion criteria are summarized in Figure 1.

We excluded CKD patients from our study because CKD is defined by markers of kidney damage or decreased glomerular filtration rate (GFR) persisting for >3 months prior to admission and was classified according to the cause, GFR, and albuminuria criteria (CGA classification) [11,12]. 

The severity of SARS-CoV-2 infection was categorized into four different types: mild, moderate, severe, or critical, according to the National Health Commission of China guidance version 7.0 (34) based on clinical symptoms, imaging findings, respiratory failure, and mechanical ventilation: “Mild cases only exhibit clinical symptoms, whereas moderate cases may include fever, respiratory symptoms, and radiological pneumonia findings. Severe cases meet all the aforementioned criteria and exhibit respiratory distress, oxygen saturation  ≤ 93% at rest, or arterial oxygen tension (PaO_2)_/fraction of inspired oxygen (FiO_2_)  ≤ 300 mmHg. Critical cases may require mechanical ventilation due to respiratory failure, shock, or other organ failures requiring intensive care unit care.” [8,9,10]. The CT scan severity score used to evaluate the extent of pulmonary involvement was based on the thin-section CT score reported by Chang et al. related to the involved area. There was a score of 0–5 for each lobe (0—none; 1—<5% of a lobe, minimal but not normal; 2—5%–25% of lobe; 3—26%–49% of lobe; 4—50%–75% of lobe; 5—>75% of the lobe), with a total possible score of 0–25 [13].

### 2.3. AKI Definition, Risk Factors, and Clinical Symptomatology

We investigated the available records for each patient for whom we excluded any renal disease of different etiology to document an increase in serum creatinine of at least 0.3 mg/dL (or 26.5 μmol/L) within 48 h or a 50% increase in serum creatinine from the baseline (creatinine ratio) within seven days according to the Acute Kidney Injury Working Group (KDIGO) [11,12]. Laboratory data were carefully analyzed at admission and during hospitalization. 

We assessed the acknowledged risk factors, such as age > 65 years, gender, and comorbidities such as neurological/psychiatric diseases, hypertension/cardiovascular diseases, diabetes, obesity, immune disease, and cancer, and coded them as categorical variables. 

We also assigned categorical variables for clinical symptoms involving different systems, such as neurological (cephalea, asthenia, anosmia, and ageusia), cardiorespiratory (cough, rhinorrhea, dyspnea, epistaxis, chest pain, and heart palpitations) or digestive system (dysphagia, vomiting, nausea, loss of appetite, diarrhea, discomfort, and abdominal pain) or for symptoms of a viral infection such as arthralgia, myalgia, and viral exanthem. Similarly, we classified the data on oxygen therapy, antiviral medication use, including lopinavir–ritonavir, favipiravir, remdesivir, anti-inflammatory, immunomodulatory, and iron chelators (hydroxychloroquine, tocilizumab, anakinra, corticosteroids, deferasirox) antibiotics (cephalosporins, carbapenems, aminoglycosides, polymyxin, cyclines, glycopeptides, fluoroquinolones, oxazolidines, macrolides, and imidazoles).

We employed multinomial variables for further analysis based on the severity of the infection (mild, moderate, severe, or critical) or the discharge status (deceased, improved, fully healed/recovered). 

Numerical variables were considered for age and no. of hospitalization days and laboratory analyses. 

### 2.4. Statistical Analysis

Continuous and categorical variables were analyzed using univariate descriptive statistics (mean, median, interquartile range (IQR) with 95% confidence intervals (CIs) and frequencies/percentages, respectively). To assess the normal distribution of the data, we utilized both Kolmogorov–Smirnov and Shapiro–Wilk tests. 

We performed the Mann–Whitney test to compare continuous variables as the distributions were skewed. In contrast, we used the chi-square test, Fisher’s exact test, or Cramer’s V for categorical variables.

The reported *p*-values were two-sided, with an alpha level of 5. We performed the computations using SPSS version 21 (IBM). 

Then, a CART (Classification and Regression Trees) algorithm was employed to identify potential predictors of the outcome (study cohort).

Decision trees serve as a machine learning method for categorizing data into different groups and uncovering concealed patterns within the dataset, as documented in prior investigations [14]. The resulting diagram forms a tree-like model of decisions, considering the target variable, which in our case is the previously presented overall outcome. It is constructed in a reverse manner, starting from the top with the root and branching out until the splitting ceases, interconnecting all predictors to predict the overall outcome. Branching occurs based on a condition (internal node) placed on the predictor variable to further divide it into branches and reach a decision. The termination point of a branch is referred to as a “leaf,” representing the decision or the child node. The criteria for stopping tree growth can be determined by either the most extended path length from the root to a child node or by selecting the minimum number of training inputs for each parent node and child node (6 and 4 in our model).

The CART algorithm functions as a dichotomous decision tree that employs the Gini index and entropy principle to divide data according to predictive variables and node purity, advancing from the parent to offspring nodes. The optimal solution substantially enhances node purity and is selected from a range of plausible partitioning paths. This iterative process continues until the stipulated cessation criteria are fulfilled, or any further decrease in node impurity becomes unattainable. The central objective revolves around pinpointing the supreme partition point (threshold value) for a predictive variable, amplifying the partitioning criteria in line with the Gini index, towing measure for impurity with categorical variables, or the LSD (Least Squares Deviation) measure for impurities in the context of continuous variables. The algorithm then ascertains the best fitting node division by selecting the predictive factor that maximizes the partitioning criterion, culminating in the most notable reduction in node impurity. This procedure is reiterated for every child node, ceasing only when further enhancements are unattainable or when predetermined halting criteria are satisfied. Typically, the user determines the minimum threshold for advancement and is often positioned at 0.0001. The CART decision tree demonstrated adaptability in accommodating diverse data types and distributions, resilience against anomalies, and skilled handling of missing data by means of alternate divisions through its fully automated mechanism. Hence, we employed the CART decision tree method because it offers the significant advantage of being nonparametric, thus making it adaptable to various types of input data. Furthermore, as previously suggested, is robust to outliers in datasets since the partitioning is based on sample proportions within specified ranges rather than absolute values [14,15,16,17]. Additionally, by utilizing surrogates for data splitting, CART appropriately handled missing data, allowing us to generate robust models even in scenarios where the level of missingness reached up to 23% for the target variable in the study group.

The variables plugged into the model for the training set were selected such that those with more than 23% missing values were excluded from the analysis, and those with missing values were considered missing, not at random (MNAR) [9,14,15,16]. We used the algorithm with statistically significant variables in the univariate analysis:Categorical (ordinal): COVID-19 clinical forms (infection severity) categorized as mild, moderate, severe, or critical according to clinical symptoms and status on discharge (cured, improved, or deceased).Dichotomous variables such as gender ICU admission, respiratory support, dyspnea sweating, myalgia, cephalea, asthenia, anosmia, ageusia abdominal pain, comorbidities such as diabetes, neurological impairment, stroke, other neurological comorbidities, hematological disorders treatment with favipiravir, remdesivir, hydroxychloroquine, tocilizumab, anakinra, corticosteroids, deferasirox, antibiotics (carbapenems, aminoglycosides, polymyxin, cyclines, fluoroquinolones, oxazolidines, macrolides, imidazoles).Continuous variables:

No. of hospitalization days, potassium levels (K^+^-levels), leukocytes, NLR (neutrophils/lymphocytes ratio), ESR, C-reactive protein (CRP), procalcitonin, D-dimer, Ferritin, LDH, creatine kinase (CPK), fibrinogen, AST, ALT, initial serum urea, creatinine concentration (at admittance), creatinine and urea levels during hospitalization. 

The accuracy of the prediction model was measured using a 10-fold cross-validation method. The set was split recurrently according to the parameter “k = 10” the split. Then, we computed the AUC for the ROC curve (AUC) and tested the accuracy.

## 3. Results

All 466 patients were evenly distributed by gender and age in both groups (Table 1 and Table 2): 104 males in the non-AKI group and 121 in the AKI group. The ages ranged between a minimum of 32 and a maximum of 93 years, with a mean of 66.97 for non-AKI cases versus a minimum of 28 years, maximum of 90 years and a mean of 67.55 in the AKI group. There was a statistically significant difference between the clinical form distribution in the two groups, with fewer mild (2 vs. 5) and moderate (54 vs. 133) cases in the AKI vs. non-AKI group. Conversely, as expected, more severe and critical cases were recorded in the AKI cohort (116 vs. 92 and 60 vs. 3, respectively) (Table 1). On the same note, 124 of the AKI patients were monitored in the ICU compared to only eight cases in the non-AKI group (*p*-value < 0.001) (Table 1), and 172 AKI patients needed oxygen therapy versus 95 non-AKI patients (*p*-value < 0.001). Then, from the AKI group, 7 out of 233 patients required dialysis (3%). In the non-AKI group, there were 4 deaths (1.71%) vs. 120 deaths in AKI cases (51.5%) (*p*-value < 0.001). Of 466 patients, 40 cases (8.5%) were discharged with improved status and 302 (68.66%) with cured status. 

The mean number of hospitalization days was significantly lower in the study group (*p* = 0.010), given that the death toll was higher (Table 1).

Then, regarding the renal function, we noted significant differences between the control group and the study group with higher levels of initial serum urea and serum creatinine concentrations:

In the study group, we further computed the mean, median, and SD values for the serum urea and serum creatinine maximum concentrations as well as the ratio of initial urea relative to the maximum value and the ratio of initial creatinine relative to the maximum value (Table 3). The maximum serum urea and creatinine concentrations were the highest levels recorded for each parameter in each patient. They were significantly increased in the subgroup of AKI patients with kidney injury onset after the first week (*p* < 0.05).

We noted statistically significant differences between groups related to tissue lesions (LDH as a global marker of injury), particularly at the pulmonary level (imaging changes-chest CT severity score), hepatic (AST, ALT), and muscular (creatinine phosphokinase) level. In addition, there was an exacerbated procoagulant and inflammatory profile in the study group: higher levels of ferritin, fibrinogen, IL-6, CRP, leukocyte count, and neutrophil/lymphocyte ratio were noticed (Table 4). 

Regarding comorbidities, diabetes mellitus was documented in 49 non-AKI patients vs. 75 AKI patients (*p* = 0.006) (Table A5). We noted a tendency towards statistical significance (*p* = 0.097) in the nutritional status with more cases in the AKI group but for the obese category (94 vs. 72) and a converse pattern in the non-AKI patients (pre-obesity in 80 non-AKI versus 72 AKI). Hypertension (145 non-AKI vs. 161 AKI patients) and other cardiovascular diseases (107 vs. 115) were balanced between the groups (*p* > 0.05). However, there were significant differences in hematological disorders, with fourteen cases in the study group and only five cases in the control group (*p* = 0.034).

Fever was present in 117 patients in both groups, cough in 126 non-AKI vs. 115 AKI patients (*p* = 0.331), and chest pain in 28 vs. 36 AKI patients (*p* = 0.273) (Table A1). Statistically significant differences were associated with dyspnea, twice as frequently in the AKI group (*p* < 0.001), sweating (18 non-AKI vs. 44 AKI, *p* < 0.001), and myalgia (48 vs. 31, *p* = 0.038) (Table A1 and Table A2). Neurological symptomatology was also significantly present in patients with AKI: anosmia (*p* = 0.032) and ageusia (*p* = 0.032) (Table A3). Nevertheless, headache was a more common symptom in non-AKI patients (80 vs. 54, *p* = 0.008). Digestive symptomatology was balanced between the groups, that is, dysphagia, vomiting, nausea, inappetence, and diarrhea (*p* > 0.05), except for abdominal pain recorded in 17 non-AKI vs. 6 AKI cases (*p* = 0.019) (Table A4). 

Antiviral therapy was administered according to the national protocols in pandemic waves, with a more predominant use of favipiravir than remdesivir in the AKI group (35 vs. 63 for favipiravir, *p* = 0.004; 27 vs. 53 for remdesivir, *p* = 0.001) (Table A6). The administration criteria were based on case severity and were related to the degree of hepatic and renal damage. Immunomodulatory medications were used more frequently in the study group: tocilizumab 4 vs. 21 in AKI patients (*p* < 0.001), anakinra (0 vs. 10, *p* = 0.001), corticosteroids (163 vs. 203, *p* < 0.001), and iron chelators (5 vs. 17, *p* < 0.001) (Table A7). 

The statistical significance for giving specific classes of antibiotics (carbapenems, aminoglycosides, polymyxins, oxazolidinones, cycline, fluoroquinolone, imidazole) and antifungals, especially in the AKI group, was related to the severity of cases associated with sepsis, pneumonia, or urinary tract infections, *Clostridioides difficile* infection especially in critical patients, and hospitalized for prolonged periods in the intensive care unit.

Our CART algorithm approach yielded decision paths suggesting three categories at risk for AKI progression: low-risk (0–40%), medium-risk (40–80%), and high-risk categories (>80%). Specific cut-off points emerged for each parameter as independent variables in the decision tree with significance to the outcome (Figure 2).


**High Risk AKI group (>80%):**
Overall, 133 AKI patients with procalcitonin > 0.045 ng/mL, initial creatinine > 1.095 mg/dL, and initial urea > 38.50 mg/dL, of which 131 (98.5%) and only 2 non-AKI patients (1.5%);Additionally, 50 patients with procalcitonin > 0.320 ng/mL, initial creatinine ≤ 1.095 mg/dL, and ICU admittance (YES), of which 50 (100%) were AKI and 0 (0%) were non-AKI;Additionally, 11 patients with procalcitonin ≤ 0.045 ng/mL and initial creatinine > 1.265 mg/dL, of which 11 (100%) were AKI and 0 (0%) were non-AKI;Five patients with procalcitonin ≤ 0.045 ng/mL, initial creatinine > 1.095 mg/dL but ≤1.265 mg/dL, and ferritin ≤ 247.45 mg/mL, of which four (80%) were AKI, and one (20%) was non-AKI.


All seven AKI patients necessitating RRT in the form of dialysis were recorded in this class.


**Medium Risk AKI group (40–80%):**
Overall, 22 patients with procalcitonin > 0.045 ng/mL, initial creatinine > 1.095 mg/mL, and initial urea ≤ 38.5 mg/mL, of which 14 (63.3%) were AKI, and 8 (36.4%) were non-AKI;Additionally, 13 patients with procalcitonin ≤ 0.320 ng/mL, initial creatinine > 0.985 mg/mL but ≤1.095 mg/mL, and LDH > 376.500 U/L, of which 8 (61.5%) were AKI, and 5 (38.5%) were non-AKI;Seven patients with procalcitonin > 0.320 ng/mL, initial creatinine ≤ 1.095 mg/mL, and ICU admittance (NO), of which three (42.9%) were AKI and four (57.1%) were non-AKI;Thirteen patients with procalcitonin ≤ 0.320 ng/mL, initial creatinine > 0.985 mg/mL but ≤1.095 mg/mL, and LDH > 376.500 U/L, of which eight (61.5%) were AKI, and five (38.5%) were non-AKI;Eight patients with procalcitonin ≤ 0.320 ng/mL, initial creatinine > 0.985 mg/dL but ≤1.095 mg/dL, LDH ≤ 376.500 U/L and initial urea > 44.50 mg/dL, of which four (50%) were AKI, and four (50%) were non-AKI.



**Low Risk AKI (0–40%):**
Nine patients with procalcitonin ≤ 0.045 ng/mL initial creatinine > 1.095 mg/dL, but ≤1.265 mg/dL and ferritin > 247.45 mg/mL, of which one (11.1%) was AKI, and eight (88.9%) were non-AKI;Additionally, 185 patients with procalcitonin ≤ 0.320 ng/mL and initial creatinine ≤ 0.985 mg/mL, of which 7 (3.8%) were AKI, and 178 (96.2%) were non-AKI;Finally, 23 patients with initial creatinine > 0.985 mg/mL but ≤ 1.095 mg/mL and procalcitonin ≤ 0.320 ng/mL, LDH ≤ 376.500 U/L and initial urea ≤ 44.50 mg/dL, of which 0 (0%) were AKI, and 23 (100%) were non-AKI.


The CART algorithm performance was notable, with an overall performance of 93.3% and 93.1% for controls, and 93.6% for the study group (Table A9—Appendix A).

## 4. Discussion

In this retrospective study of COVID-19 patients, we compared demographical, clinical, and biological biomarkers as risk factors associated with AKI and non-AKI outcomes.

### 4.1. Epidemiological and Clinical Considerations 

A review of the current literature on AKI prevalence in COVID-19 patients showed highly inhomogeneous data ranging from 0.5% [18] to 5.1% [1], 12.7% [19], 29.6% [20], 36.6% [21,22], and 85% [4]. Death rates were much higher in unrecovered AKI cases in the presence of critical illness with pulmonary involvement [23]. We documented a significantly higher death rate in AKI cases (51.5% vs. 1.71%), which is consistent with Cheng Y et al. 2020 and Case J et al. 2013 [1,2].

Amongst comorbidities or conditions associated with SARS-CoV-2 severe forms, cardiovascular diseases (especially arterial hypertension), diabetes mellitus, and hepatic and renal insufficiency stand out, suggesting an unfavorable outcome [1]. However, regarding the aforementioned risk factors related to AKI categories, there was increased heterogeneity in the reported data. For example, a systematic review and meta-analysis performed by Lim et al. 2020 in 15 studies comprising 3615 patients revealed no association between AKI category and age, sex, cardiovascular diseases (including arterial hypertension), respiratory and renal chronic diseases, and diabetes [7]. Conversely, Sabaghian et al. 2022 documented diabetes, arterial hypertension, hyperlipidemia, and chronic renal disease in 114 COVID-19 patients [3]. 

As previously mentioned, many patients with SARS-CoV-2 infection develop AKI, and some of the most severe cases require renal replacement therapy (RRT). Among patients who develop acute respiratory distress syndrome from all causes, 10–14% of the patients who develop severe forms of AKI require RRT [24], which is an independent predictor of mortality and is associated with poor outcomes among patients with SARS-CoV-2 infection [25]. Stevens et al. reported data from an analysis of 510 patients with SARS-CoV-2 infection admitted to the ICU; 115 (23%) patients received RRT for AKI. By the end of their follow-up period, of the 115 patients who received RRT, 59 (51%) died, 47 patients (41%) had evidence of kidney recovery, and 9 patients (8%) still required RRT. In conclusion, their study demonstrated a high incidence (23%) and peak prevalence (29%) of severe AKI required RRT. Their results indicated a high rate of recovery of kidney function among survivors (84%) [26].

Tan et al. published the results of a retrospective multi-center observational cohort study of 12,891 patients with SARS-CoV-2 infection across five countries. Almost half of the patients reported AKI (49.5%). COVID-19-associated AKI has been linked to poorer long-term kidney function recovery and higher all-cause mortality rates, even among survivors who passed the 30-day mark following their initial AKI event. Advanced age, severe forms of SARS-CoV-2 infection, severe AKI, and ischemic heart disease are among the factors associated with more significant mortality. Severe AKI has also been found to be significantly associated with worse kidney function recovery outcomes, whereas the use of remdesivir has been associated with improved recovery in select patients. Patients without chronic kidney disease at baseline may experience kidney function impairment due to advanced age, male gender, severe AKI, and hypertension. Prior exposure to ACE inhibitors, angiotensin-II receptor blockers, or remdesivir has been found to have reno-protective effects [27].

We report statistical significance for diabetes only in 233 patients with AKI compared with balanced controls. Therefore, from our perspective, it seems evident that there is a need for further prospective studies on larger COVID-19 cohorts to properly assess the impact of associated conditions on AKI development.

Dyspnea, sweating, and myalgia were statistically significant symptoms in both the AKI and non-AKI groups. Moreover, the presence of dyspnea at admission, which is twice as frequent in AKI patients compared to non-AKI patients, and the need for oxygen supplementation and ICU monitoring was decisive for an unfavorable outcome in the AKI group. Most neurological symptoms and stroke were more prevalent in patients with AKI. In non-AKI cases, cephalea, anosmia, ageusia, and abdominal pain were observed more frequently. Future studies should these phenotypes.

Even though the majority of reported studies do not strictly refer to AKI as an independent mortality risk factor for COVID-19, and the number of enrolled subjects is too small relative to the COVID-19 phenomena dimension, in the previously referred meta-analysis, Lim et al. presented similar results with ours in terms of high mortality rate among AKI patients with severe clinical forms (10 times higher) along with a higher ICU monitoring rate [7]. 

### 4.2. Pathophysiological Mechanisms for Renal Injury

Our study noted renal damage in SARS-CoV-2 infection during alpha- and delta- variant circulation. The pathophysiological mechanisms vary according to the time of occurrence. In the first week, we stressed key factors such as dehydration in a febrile context, lack of oral intake, or losses through gastrointestinal disturbances (vomiting and diarrhea), while in the second week, the immune mechanisms, coagulation disorders or sepsis played a crucial role (Table 1 and Table 2).

We agree with the other authors’ studies [28,29]. The current literature suggests that direct renal damage seems to be the consequence of SARS-CoV-2 infection on podocytes in cases where significant glomerular damage is associated with significant proteinuria [18,30]. SARS-CoV-2 presence in the kidney has a specific inflammatory response with tubulointerstitial macrophage infiltration and tubular complement deposition [31]. At the proximal tubular level, the SARS-CoV-2 concentration was 100 times higher than the lung levels, exerting a direct tubular effect [32]. The presence of SARS-CoV-2-like particles has also been confirmed in patients with focal segmental glomerulosclerosis and acute tubular necrosis [33,34]. Renal and myocardial damage and their negative influence during SARS-CoV-2 viremia seem to be explained by higher levels of ACE-2 receptors in the kidney (4%) and myocardium (7.5%) compared to in pneumocytes (2%) [35]. Another possible mechanism involved in renal failure is rhabdomyolysis, with myoglobin release negatively impacting the outcome through intrarenal renal vasoconstriction, tubular ischemia, and tubular obstruction [36].

The COVID-19 pandemic has also revealed reports of collapsing glomerulopathy similar to those documented during the HIV epidemic. Regarding the clinical background of these patients, they presented AKI and nephrotic-range proteinuria. Histopathological examination of the lesions was similar to that observed in other forms of collapsing glomerulopathy (segmental/global collapse of the glomerular capillary tuft, hypertrophy in association with hyperplasia of the podocytes and parietal epithelial cells, and protein resorption droplets within hyperplastic glomerular epithelial cells). This type of nephropathy should be distinguished from most cases of AKI in SARS-CoV-2 infection, which are characterized by acute tubular injury. Tubulointerstitial lesions typically associated with apolipoprotein L1 (APOL1)-related diseases include microcystic tubular dilatation and tubular injury. COVID-19-associated nephropathy (COVAN) can affect individuals in some regions of the world. This entity should be particularly suspected in cases of African descent patients who present with SARS-CoV-2 infection, AKI, and nephrotic-range proteinuria. Individuals of African ancestry present an increased risk of chronic kidney disease (CKD) and kidney failure in the context of the presence of a polymorphism in the APOL1 gene [37,38,39]. Patients hospitalized with SARS-CoV-2 infection are an increased risk of developing long-term adverse health outcomes. Part of these patients will present long-term kidney dysfunction. Identifying this category of patients at high risk for adverse kidney events is important in terms of a nephrology follow-up. In a prospective cohort of 576 patients hospitalized with COVID-19, Menez et al. reported that the plasma biomarkers soluble tumor necrosis factor receptor 1 (sTNFR1) and sTNFR2 measured in admitted patients with COVID-19 were associated with a greater risk of adverse kidney outcome, which are strong predictors of these events. In addition, patients hospitalized with COVID-19 and increased urinary biomarkers, such as neutrophil gelatinase-associated lipocalin, monocyte chemoattractant protein, and kidney injury molecule 1, are at high risk of adverse kidney outcomes and death [40,41].

Therapy with nephrotoxic potential mainly involves the administration of 1. antivirals (remdesivir, with potential mitochondrial toxicity, lopinavir/ritonavir, responsible for acute tubular necrosis), 2. antibiotics: aminoglycosides, glycopeptides, and beta-lactams may be responsible for tubular injury, including their proximal accumulation [42]; 3. monoclonal antibodies: adalimumab, by increasing capillary permeability with decreased circulating blood volume; 4. vasoactive medications: adrenaline, noradrenaline, dopamine, with the risk of pre-renal azotemia; and 5. antifungal medication: amphotericin B.

Given the multiple overlapping risk factors that might trigger AKI, it was difficult to conclude the role of antiviral, immunomodulatory, anticoagulant, antibiotic, or antifungal medications as key triggers in our study. The most probable AKI is the consequence of renal damage secondary to SARS-CoV2 viremia and associated comorbidities and conditions. In addition, it should be noted that the CART algorithm’s ability to explore hidden links and patterns within the given data did not select any of the aforementioned nephrotoxic drugs as predictors of AKI outcome. 

### 4.3. Inflammation Procoagulant Status and Renal Injury

The second week after the onset of SARS-CoV-2 infection is currently regarded as crucial from an inflammatory status perspective, given the increased levels of proinflammatory cytokines. IL-1, IL-6, IL-12, and IFN-gamma negatively impact pneumocytes and renal cells, favoring their apoptosis-primarily through the action of IFN-gamma [43,44]. Moreover, IL-6 values (highly correlated with C-reactive protein), along with IL-8 and IL-10, are highly predictive both for SARS-CoV-2 infection severity (86.4%, 95% CI:72.4–94.8), with a specificity of 94.7%, [45] and for the need of case monitoring in intensive care. Furthermore, SARS-CoV-2 infection creates a procoagulant status [46,47], with elevated D-dimer, fibrinogen, and fibrin levels, especially in people who do not survive COVID-19 [48]. According to Xiong M et al. 2020, 71.4% of those who died had consumptive coagulopathy (CID) [49]. Thrombotic microangiopathy occurring in a CID setting was associated with acute kidney injury [50]. The aftermath of the inflammatory response is consumption coagulopathy, hypoxia, and renal microthrombosis [51,52]. These seem to be the most convincing mechanisms that elicit renal injury in COVID-19 patients. Siguret et al. documented the presence of antiphospholipid antibodies in 85% of critical COVID-19 cases, possibly attributable to cytokine storm and dysimmunity, and a 12% prevalence of anticardiolipin/anti-β_2_-glycoprotein-I antibodies in these patients [51]. In our cohort, we noted that relative to the onset of renal failure, patients developing AKI in the first week of illness had a favorable clinical status towards recovery, with lower procalcitonin and CRP values. Patients developing AKI in the second week of the disease had much higher procalcitonin, CRP, and D-dimer levels, suggesting a greater severity in the septic or cytokine storm context. Our findings align with those of other authors, who reported the association of AKI with high ferritin values, CRP, and severe pulmonary changes [30] or with high levels of LDH, procalcitonin, or ASAT and lymphopenia [46]. The CART algorithm revealed a specific cut-off point for procalcitonin (>0.32 ng/mL in combination with initial creatinine >1.1 mg/dL or >0.045 mg/dL or with initial creatinine > 1.1 mg/dL and initial urea > 38.5 mg/dL) to stand out as an important reference value for patients at a high risk to develop AKI. Procalcitonin levels < 0.045 ng/mL, initial creatinine levels > 1.1 mg/dL, and ferritin < 247.5 mg/mL also placed the patients in the high-risk category. An important observation to be highlighted is the absence of PCR in the CART selected model and the fact that LDH > 376.5 U/L and elevated procalcitonin levels, still ≤ 0.320 ng/mL, would classify the patients in the medium risk category. It remains unclear how LDH performance is an indicator that would affect progression to AKI [53]. Recent reports cast doubts about the predictive capacity of LDH for mortality, given the decrease in its cut-off levels documented by Hernandez et al. in a retrospective study of 843 patients. High LDH serum levels in the inflammatory context indicate cell membrane disruption, platelet, and angiogenesis activation [54]. Inflammation is crucial for multiple organ dysfunctions with exacerbated cytokine production, especially in the lungs and kidneys. Our results demonstrate elevated inflammatory and tissue injury biomarkers and reinforce this hypothesis; however, pinpointing the exact AKI mechanisms and deciphering the kidney’s roles in multiple organ dysfunction calls for further research. 

Wang et al. introduced a robust risk assessment framework employing multivariate logistic regression analysis but on fewer key variables such as oxygen saturation (SaO_2_), procalcitonin (PCT), and blood urea nitrogen (BUN). Their model proposed risk categories based on cumulative scores, discerning low risk (score = 0), medium risk (score = 2), and high risk (score = 3) [55]. For the low risk scenario, SaO_2_ fell below 97.5 (score = 0), accompanied by a PCT value below 0.1045 (Score = 0) and a BUN level below 5.565 (score = 0). For the medium-risk scenario, any of the three variables should have had a score of 1 meaning > than its cut-off value. In contrast, the high-risk category emerged with a total score of 3, with all the variables scoring 1 for each. In contrast, our approach evaluated a broader spectrum of parameters for Acute Kidney Injury (AKI) grouped based on association rules. Procalcitonin > 0.320 ng/mL signified high risk, 0.045–0.320 ng/mL indicated medium risk, and <0.045 ng/mL suggested low risk. An elevated initial creatinine (>1.095 mg/dL) implied high risk, 0.985–1.095 mg/dL reflected medium/high risk, and ≤0.985 mg/dL signified low risk. Elevated urea (>38.50 mg/dL and >44.50 mg/dL) denoted high/medium risk, while ≤38.50 mg/dL was associated with medium/low risk. LDH > 376.500 U/L was related to medium/high risk, and ≤376.500 U/L indicated low risk. Ferritin > 247.45 mg/mL suggested low risk. Notably, our approach blended more variables with finer cut-off points in association rules, enhancing the risk evaluation’s granularity and guiding tailored clinical prognosis in more detail.; however, our model did not select age, diabetes, and hypertension compared to Palomba et al.’s risk score on a larger cohort [56]. However, their model did not evaluate PCT and other inflammatory biomarkers, while our design was confronted with data missing from the ICU records concerning angiotensin-converting enzyme inhibitors (ACEi), angiotensin receptor blockers (ARBs), and the need for vasopressor and mechanical ventilation. We acknowledge this aspect as a limitation that should be addressed in future studies with larger cohorts.

### 4.4. Study Limitations

To resume, we assessed in AKI versus non-AKI COVID-19 patients the most referred comorbidities, clinical symptoms, serum biomarkers, and treatment types in the literature. However, certain limitations of our study should be considered. Although our sample size was significantly related to the number of total admissions in our tertiary hospital, as other authors have stated, we agree with Peng et al. that AKI incidence, in general, might be underestimated in the literature as a consequence of data collection in different research centers [57]. Given the retrospective nature of our study, AKI onset was difficult to detect before or after admission. Therefore, we considered the first, second, and third weeks after admission as appropriate periods to assess AKI onset under these circumstances. Additionally, our study lacked follow-up data regarding the percentage of discharged patients who experienced progression to chronic kidney disease. This limitation restricted our ability to assess the long-term outcomes and determine the extent of chronic kidney disease development in the studied population.

On the other hand, the percentage of severe and critical cases was 58.15% of the sample population; hence, this aspect should be considered when it comes to external validation of our results. Another limitation was related to the lack of AKI stadialization and analysis of other organ insufficiencies (i.e., hepatic and cardiac).

Finally, certain limitations regarding the CART model should also be addressed as this was a retrospective study based on the data retrieved from 466 patient records. Consequently, as Peng et al. suggested, residual confounding may exist in our approach [44]. Our results are notable in terms of the model performance. However, the external validity of larger cohorts should be further explored [8,9]. A more significant number of patients would provide a more precise AKI prognosis, especially if we consider AKI classification with three stages as the outcome variable in our CART analysis approach. Hence, using the strata based on the KDIGO classification with three stages would further entail enhanced forecasting patterns and specific cut-off points for splitting based on these categories: AKI stage 1 is serum creatinine 1.5–1.9 times baseline or ≥0.3 mg/dL increment or urine output less than 0.5 mL/kg/h for 6–12 h; AKI stage 2 is serum creatinine 2.0–2.9 times baseline or urine output less than 0.5 mL/kg/h for more than 12 h; and AKI stage 3 is serum creatinine more than 3.0 times baseline or increment in serum creatinine to ≥4.0 mg/dL.

## 5. Conclusions

Our study revealed the association between particular risk factors and AKI progression in COVID-19 patients. Diabetes, the presence of dyspnea on admission, the need for supplemental oxygen, and admission to the intensive care unit all had a crucial role in unfavorable outcomes with a death rate of more than 50%. Important imaging studies (CT scan severity score) and changes in specific biomarkers levels, especially ferritin and C-reactive protein, were also noted and should be further investigated in correlation with the pathophysiological mechanisms of AKI progression in COVID-19 patients. Further research should explore and decipher the role of the kidney in SARS-CoV-2 infections either as an aftermath of a severe infection triggering multiple system organ failure or as a pointer in the scoring system that assesses the performance of several organ systems in the body (sequential organ failure assessment method).

## Figures and Tables

**Figure 1 healthcare-11-02402-f001:**
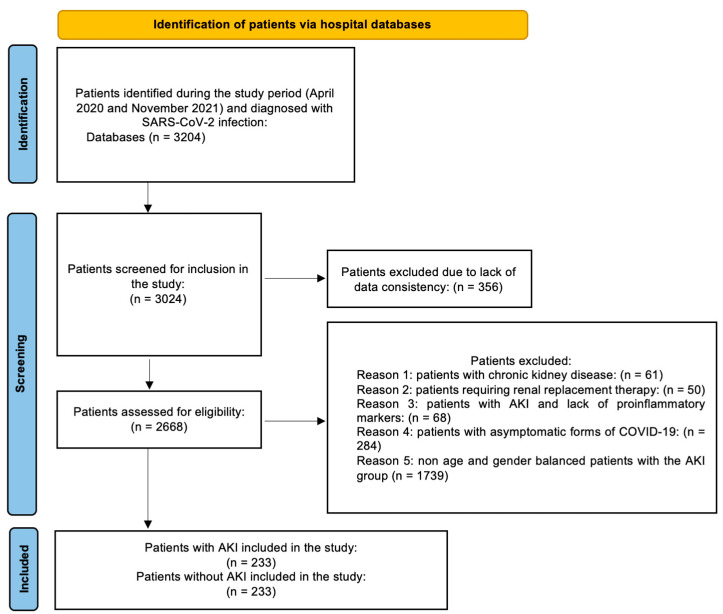
Flow diagram of identification of patients.

**Figure 2 healthcare-11-02402-f002:**
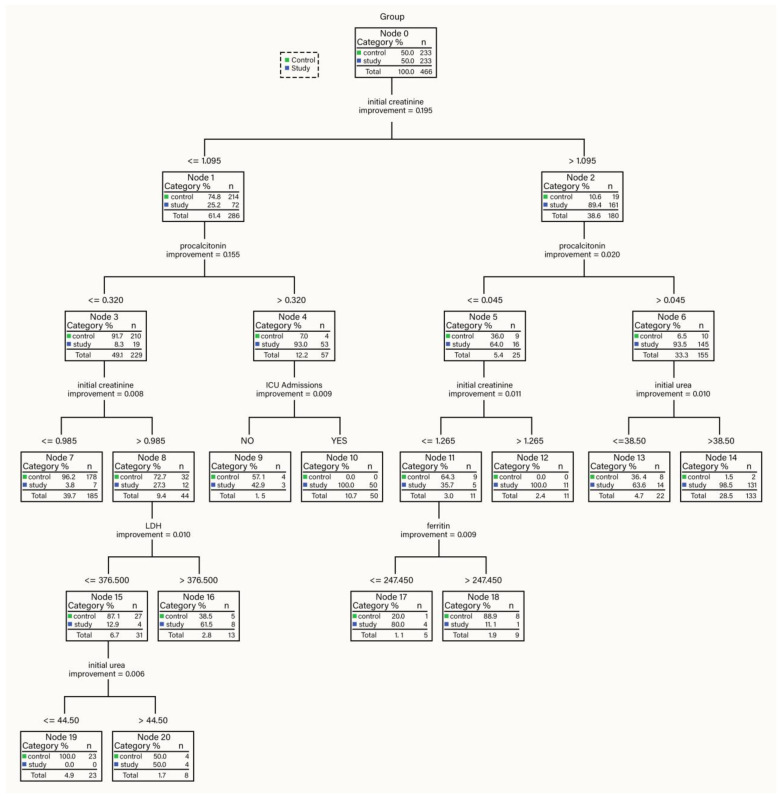
Decision path for each risk group.

**Table 1 healthcare-11-02402-t001:** Categories for clinical forms, respiratory support, ICU admission, and discharge status.

Parameter	Groups			Total (n)	OR LR	*p*-Value ^B^
	Categories	Non-AKI Control (n)	AKI Study (n)			
Gender	Male	104	121	225	LR = 2.486	0.115
	Female	129	112	241	OR = 0.746	
	Total	233	233	466	95% CI (0.518, 1.074)	
COVID-19 clinical forms ^B^	Mild	5	2	7	LR = 101.762	<0.001
	Moderate	133	54	187		
	Severe	92	116	208		
	Critical	3	60	63		
	Total	233	232	465		
Status on discharge ^B^	Cured	210	92	302	LR = 184.016	<0.001
	Improved	19	21	40		
	Deceased	4	120	124		
	Total	233	233	466		
ICU admission	No	225	109	334	LR = 163.771	<0.001
	Yes	8	124	132	OR = 32	
	Total	233	233	466	95% CI (15.104, 67.77)	
Respiratory support	No	138	61	199	LR = 53.11	<0.001
	Yes	95	172	267	OR = 4.1	
	Total	233	233	466	95% CI (2.768, 6.061)	

N—number of cases, LR—likelihood ratio, OR—Odd ratio (95%CI), Chi-Square, Cramer’s V test ^B^.

**Table 2 healthcare-11-02402-t002:** Renal function in the study group versus the control group.

Variables	Control	Study	*p*-Value ^A^
M, SD (95% CI), Min, Max, IQR, Median			
Age (years)	67 ± 11.2	67.6 ± 11.2	0.516
	(65.54–68.41)	(66.11–69.00)	
	32–93	28–90	
	60–75 (67)	60–76 (69)	
No. of hospitalization days	16 ± 6.1	15.6 ± 9.6	0.010
	(15.22–16.80)	(14.33–16.82)	
	2–42	1–77	
	12–18 (15)	10–19 (14)	
Initial serum urea concentration (mg/dL)	24.1 ± 11.6	66.6 ± 42.5	<0.001
	(32.547–35.563)	(61.122–72.111)	
	0.8–99	0.0–280	
	26–40 (33)	41.3–76 (55)	
Initial serum creatinine concentration (mg/dL)	1 ± 2	1.4 ± 1	<0.001
	(0.741–1.255)	(1.280–1.510)	
	0.52–31	0.59–8.31	
	0.8–1 (0.84)	1–1.5 (1.24)	
Blood sodium level (mEq/L)	136 ± 20.5	139.4 ± 910	0.504
	(133.352–138.639)	(138.086–140.695)	
	0–149	41–183	
	137–141 (139)	135–143 (139)	
Blood potassium level (mEq/L)	4.058 ± 0.52	4.32 ± 1.01	0.044
	(2.03–5.67)	(4.18–4.45)	
	2.03–5.67	2.25–8.85	
	3.75–4.42 (4.065)	3.64–4.77 (4.16)	

M—mean, SD—standard deviation, Min—minimum, Max—maximum, IQR—interquartile range, Median, Mann–Whitney ^A^.

**Table 3 healthcare-11-02402-t003:** Study cohort maximum serum concentrations for urea and creatinine based on AKI onset categories.

Variables	Study Cohort		*p*-Value ^A^
	AKI Onset (1st Week)	AKI Onset > 1st Week	
Abnormal initial serum urea concentration (mg/dL)	54.00	57.00	0.051
Max. serum urea concentration (mg/dL)	90.00	118.00	<0.001
Initial serum urea concentration ratio max. val. ^B^	1.509	1.687	0.145
Abnormal initial creatinine (mg/dL)	1.29	1.2	0.156
Maximum serum creatinine concentration (mg/dL)	1.52	2.08	0.002
Creatinine ratio max. val. ^B^	1.1735	1.5478	0.012

M—mean, SD—standard deviation (95% CI), Min—minimum, Max—maximum, IQR—interquartile range, Median, Mann–Whitney Test ^A^, urea ratio ^B^, and creatinine ratio ^B^ (the maximum value urea and creatinine values relative to the baseline).

**Table 4 healthcare-11-02402-t004:** Tissue lesions and systemic inflammatory response.

Variables	Control	Study	*p*-Value ^A^
N, M ± SD (95%CI); MIN, MAX, IQR (MEDIAN)			
Lactate Dehydrogenase (LDH)	296 ± 126.5	895 ± 1867	<0.001
	(278.965–312.736)	(545.235–1044.819)	
	0–830	106–17,348.85	
	214–358 (275)	298–735.5 (453)	
(CT) scan severity score	55	131	0.001
	14.6 ± 6	17.5 ± 6.1	
	(13.01–16.22)	(16.43–18.53)	
	0–24	0–25	
	12–19 (15)	14–22 (19)	
Creatine kinase	131 ± 182	544 ± 1498	<0.001
	(107.04–154.72)	(348.04–739.86)	
	0–1716	0–15,670	
	51–132.5 (87)	70–450 (154)	
AST (U/L)	39 ± 29	190.7 ± 848	<0.001
	(34.99–42.50)	(77.56–303.93)	
	0–274	12–7253	
	21–47 (30)	29–76.2 (46.5)	
ALT (U/L)	41.7 ± 41.1	134.7 ± 495.1	<0.001
	(36.35–46.95)	(69.10–200.37)	
	0–324	6–4367	
	20.5–45 (30)	24–72.5 (42)	
Ferritin (mg/mL)	631.4 ± 645	1426.3 ± 4118.4	<0.001
	(545.464–717.223)	(774.978–2077.682)	
	13.6–4241.6	28.2–47,724.4	
	191.1–820 (413)	244–465.4 (636.5)	
D-dimer (ng/mL)	1258.3 ± 1662.5	6634 ± 11,858.4	<0.001
	(1033.711–1482.814)	(5072.466–8195.252)	
	190–15,794	0–50,007	
	462.5–1364.5 (778)	700.5–5518.4 (1600.24)	
IL-6	12.6996 ±	561.7704 ±	<0.001
	(5.328–20.071)	(191.214–1314.754)	
	1.4–77.89	1.5–29,891	
	3–19.5 (6.56)	15.5–125 (41.905)	
C-Reactive protein (mg/L)	70.3 ± 80	140.7 ± 105.1	<0.001
	(60.036–80.651)	(127.024–154.387)	
	0–406.2	0–524.83	
	10.3–104.1 (37.18)	50.7–213 (125.5)	
Fibrinogen (mg/dL)	484.3 ± 180.4	556.2 ± 186.4	<0.001
	(460.743–508.047)	(531.469–580.893)	
	0–900	0–890.4	
	363–589.6 (455.15)	422.2–711.5 (550.6)	
ESR (mm/h)	41 ± 30	44.3 ± 32.4	0.341
	(36.98–44.89)	(39.87–48.81)	
	0–121	0–122	
	16–63.7 (34)	17–66.7 (40.5)	
Leukocytes (10^3^/μL)	7.5 ± 3.5	13.5 ± 9.2	<0.001
	(7.039–7.948)	(12.244–14.668)	
	1.72–20.72	1.40–50.48	
	5.2–8.6 (6.7)	6.7–18.7 (10.03)	
NLR	5.8 ± 6.6	18.7 ± 24.8	<0.001
	(4.982–6.704)	(15.506–22.087)	
	0–39.632	1.214–148.143	
	2.2–6 (3.8)	4.6–23.5 (8.8)	
Procalcitonin (ng/mL)	100	180	<0.001
	0.2 ± 0.7	8.7 ± 22.7	
	(0.073–0.358)	(5.324–12.007)	
	0–5.74	0–201	
	0.03–0.1(0.04)	0.2–6.7(0.965)	

N—number of cases, M—mean, SD—standard deviation, Min—minimum, Max—maximum, IQR—interquartile range, Median—mean, AST—aspartate aminotransferase; ALT—alanine aminotransferase; ESR—erythrocyte sedimentation rate; NLR—neutrophil-to-lymphocyte ratio, Mann–Whitney test ^A^.

## Data Availability

All data generated or analyzed during this study are included in the published article.

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
