# Peer review of "Our Experience with SARS-CoV-2 Infection and Acute Kidney Injury: Results from a Single-Center Retrospective Observational Study"

_healthcare, 2023, doi:10.3390/healthcare11172402_

Round 1

Reviewer 1 Report (Previous Reviewer 3)

Thank you for the  opportunity  to review  this  manuscript.

SARS-CoV-2 associated   AKI  represent  a significant  source of morbidity and  mortality  among patients  surviving  severe  COVID-19 disease, this  manuscript offers very important  information.

Minor issues  must  be addressed: 

Extent  briefly the epidemiological  data  of AKI  in  terms  of prognosis, associated  morbidity, mortality  and renal  recovery (specially in terms of progression  to CKD,  renal  recovery  and  renal replacement  therapy  dependence) and the  reported  incidence  of COVID-19 associated AKI  according  to  its  severity. This will emphasize  the  value  of  your  research. 

In the results  section, If  possible,  describe  and  compare within  the  groups  of  AKI progression:  the   time of AKI onset,  the AKI severity ( i.e.  need  for  RRT)  and  renal  outcome (i.e.  renal  recovery,  dialysis  dependance. This may  emphasize  the  utility  of  the  decision path  you  are proposing.   (I.e. within  the  high-risk/low risk  group, the proportion of patients ultimately depending  on  renal  replacement  therapy at  discharge). 

In the  discussion  section  briefly  compare  your  decision path   to other  risk-scale  for AKI described  for SARS-CoV-2 Infection, and  describe why you  did  not  take into account  variables  such as  vasopressor /mechanical  ventilation  need (i.e. not  statistically significant  differences  found,  greater chance  to  select patients that  ultimately may develop  Aki if initial  laboratories are  taken into account)

Good quality of  English Language

Author Response

Dear Reviewer,

Thank you for taking the time to review our revised manuscript reviewing our manuscript. Please find attached a revised version of our manuscript, “Our experience with SARS-CoV-2 infection and acute kidney injury. Results from a single-center retrospective observational study”.

Your insightful comments, along with those of the other reviewers, were consistently helpful in improving the quality of our paper.

We are pleased to inform you that we have carefully considered each of your comments and have made the necessary revisions to the manuscript. We have addressed your concerns in a point-by-point response.

We greatly appreciate your effort in offering invaluable suggestions to review our manuscript. Your contributions have undoubtedly made our work clearer highlighting the added value, and we are very grateful for your support.

Reviewer Comments:

Reviewer 1

Thank you for the opportunity  to review  this  manuscript.

#1. SARS-CoV-2 associated   AKI  represent  a significant  source of morbidity and  mortality  among patients  surviving  severe  COVID-19 disease, this  manuscript offers very important  information.

Answer: We highly appreciate your time and effort in reviewing our manuscript and your high-quality feedback. Your insights have significantly improved our research report along with the conveyed messages.

Minor issues  must  be addressed:

#2 Extent  briefly the epidemiological  data  of AKI  in  terms  of prognosis, associated  morbidity, mortality  and renal  recovery (specially in terms of progression  to CKD,  renal  recovery  and  renal replacement  therapy  dependence) and the  reported  incidence  of COVID-19 associated AKI  according  to  its  severity. This will emphasize the  value  of  your  research.

A: We thank the distinguished reviewer for highlighting this important aspect. We briefly extended the epidemiological data in the discussion section of the manuscript based on your suggestions. Please refer to the lines(373-397)

“As previously mentioned, many patients with SARS-CoV-2 infection develop AKI, and some of the most severe cases require renal replacement therapy (RRT). Amongst patients that develop acute respiratory distress syndrome from all causes, 10–14% of the patients develop severe forms of AKI enough to require RRT [25]. AKI has been referred as an independent predictor of mortality and has been associated with poor outcomes among patients with SARS-CoV-2 infection [26,27]. In the meta-analysis performed by Robbins-Juarez et al on 20 cohorts (13,137 mostly admitted patients) 6 studies provided an AKI severity breakdown (stages 1,2,3) with 15%(stage1), 7%(stage 2), 11%(stage 3). The reported incidence of RRT was 5 % with a range of 0.8%-14.7%[27]. In our retrospective analysis we documented 7 out of 233 AKI(3%) patients that required RRT in the form of dialysis and all of them had an unfavorable outcome. Stevens et al. reported data from a cohort of 510 patients with SARS-CoV-2 infection admitted to the ICU; of them, 115 (23%) received RRT for AKI. By the end of their follow-up period, of the 115 patients that received RRT, 59 (51%) patients died, 47 patients (41%) had evidence of kidney recovery, and nine patients (8%) still required RRT. In conclusion, their study demonstrates a high incidence (23%) and peak prevalence (29%) of severe AKI requiring RRT. Their results indicate a high rate of recovery of kidney function among survivors (84%) [28].Tan et al. published the results of a retrospective multi-center observational cohort study from 12,891 patients with SARS-CoV-2 infection across five countries. Almost half of the cases reported AKI (49.5%). COVID-19-associated AKI has been linked to poorer long-term kidney function recovery and higher all-cause mortality rates, even among survivors who have passed the 30-day mark following their initial AKI event. Advanced age, severe forms of SARS-CoV-2 infection, severe AKI, and ischemic heart disease are among the factors associated with more significant mortality. Severe AKI has also been found to be significantly linked to worsen kidney function recovery outcomes, while the use of remdesivir has been associated with improved recovery in select patients. Patients without chronic kidney disease at baseline may experience kidney function impairment due to advanced age, male gender, severe AKI, and hypertension. Neither prior exposure to ACE inhibitors or angiotensin-II receptor blockers, nor remdesivir, have been found to have any reno-protective effects [29].”

#3 In the results  section, If  possible,  describe  and  compare within  the  groups  of  AKI progression:  the   time of AKI onset,  the AKI severity ( i.e.  need  for  RRT)  and  renal  outcome (i.e.  renal  recovery,  dialysis  dependance. This may  emphasize  the  utility  of  the  decision path  you  are proposing.   (i.e. within  the  high-risk/low risk  group, the proportion of patients ultimately depending  on  renal  replacement  therapy at  discharge).

We highly appreciate the distinguished reviewer’s suggestion to add this important detail. Please note the added lines (325-326) in the Results section referring to all of the 7 patients that required dialysis and were categorized in the high-risk group. Along with these lines we also appended the table A9 to present a breakdown in a two-layer levels model for the categorical variables with the first layer(CART predicted value-control/study group), second layer (time of AKI onset) and the crosstabs (clinical_forms*status_on_discharge) with the counted cases based on this distribution. Please also note that we acknowledged as a study limitation “ the lack of AKI stadialization using the strata based on the KDIGO classification with 3 stages that would have further entailed enhanced forecasting patterns in the CART approach and specific cut-off points for splitting based on these categories: AKI stage 1 as serum creatinine 1.5–1.9 times baseline or ≥0.3 mg/dL increment or urine output less than 0.5 mL/kg/h for 6–12 h; AKI stage 2 as serum creatinine 2.0–2.9 times baseline or urine output less than 0.5 mL/kg/h for more than 12 h; AKI stage 3 as serum creatinine more than 3.0 times baseline or increment in serum creatinine to ≥4.0 mg/dL.“ (lines 564-578).

#4 In the  discussion  section  briefly  compare  your  decision path   to other  risk-scale  for AKI described  for SARS-CoV-2 Infection, and  describe why you  did  not  take into account  variables  such as  vasopressor /mechanical  ventilation  need (i.e. not  statistically significant  differences  found,  greater chance  to  select patients that  ultimately may develop  AKI if initial  laboratories are  taken into account)

A: We thank the distinguished reviewer for this important observation. We searched the current literature to contrast our stratified risk categories with alternative risk scoring systems that have been previously documented. Please note the lines(529-585)
“ Wang et al. introduced a robust risk assessment framework employing multivariate logistic regression analysis but on a fewer key variables such as oxygen saturation (SaO2), procalcitonin (PCT), and blood-urea nitrogen (BUN). Their model proposed risk categories based on cumulative scores, discerning low risk (score=0), medium risk (score=2), and high risk (score=3)[57]. For the low risk scenario SaO2 fell below 97.5 (Score = 0), accompanied by a PCT value below 0.1045 (Score = 0) and BUN level below 5.565 (Score = 0). For the medium-risk scenario any of the three variable should have had a Score of 1 meaning > than its cut-off value. In contrast, the high-risk category emerged with a total score of 3, with all the variables with a score of 1 for each. In contrast, our approach evaluated a wider spectrum of parameters for Acute Kidney Injury (AKI) grouped together based on association rules. Procalcitonin > 0.320 ng/ml signified high risk, 0.045-0.320 ng/ml indicated medium risk, and < 0.045 ng/ml suggested low risk. Elevated initial creatinine (>1.095 mg/dl) implied high risk, 0.985-1.095 mg/dl reflected medium/high risk, and ≤ 0.985 mg/dl signified low risk. Elevated urea (>38.50 mg/dl and >44.50 mg/dl) denoted high/medium risk, while ≤ 38.50 mg/dl associated with medium/low risk. LDH > 376.500 U/L related to medium/high risk, and ≤ 376.500 U/L indicated low risk. Ferritin > 247.45 mg/ml suggested low risk. Notably, our approach blended more variables with finer cut-off points in association rules, enhancing the granularity of risk evaluation and thereby guiding tailored clinical prognosis in more details. Nevertheless, our model didn’t select age, diabetes, and hypertension compared to Palomba’s et al risk score on a larger cohort[58]. Their model didn’t evaluate PCT and other inflammatory biomarkers while our design confronted with data missing from the ICU records with regard to angiotensin converting enzyme inhibitors (ACEi), angiotensin receptor blockers (ARBs) and the need of vasopressor and mechanical ventilation. We acknowledge this aspect as a limitation that should be addressed in future studies on larger cohorts.

We hope that the revised form of the manuscript and our accompanying responses properly addressed all the raised issues making our manuscript suitable and ready for publication in Healthcare. We shall look forward to hearing from you at your earliest convenience.

With our best regards,

Sincerely yours,

Victoria Birlutiu, Prof. Habil. M.D. Ph.D

Bogdan Neamtu, Associate Professor, M.D. Ph.D.

Rares Mircea Birlutiu, M.D. Ph.D.

Reviewer 2 Report (New Reviewer)

This is a research about covid and AKI. Much have been published already in terms of AKI covid. The positives of the study includes using machine learning for risk prediction, which can be adopted in the future. 

Suggest:

1. Add literature about COVAN, ApoL1, collapsing GN and Covid

2. Menez at al have included significant biomarkers differences in Covid aki vs regular AKI. 

3. Line 25 in abstract not clear

4. Would describe more about CART in the introduction and abstract. Otherwise, most of the study is already well known. 

5. Line 40- not clear

6. Line 43 Acute renal impairment- in consistent use of definition

7. Is there any information about baseline proteinuria and also data on albuminuria

8. One of the most common menifestations of covid and kidneys is hematuria, nothing is mentioned in terms of long term prognosis. 4 years into pandemic, i think it would provide more useful information

9. would provide more information about Sibiv Hospital, how big, catchment area, referral centers, how many bed ICU and inpatient services

10. The eligibility criteria isnot clear

11. What is the importance of creatinine ratio

12. Lines 90-95 not clear

Author Response

Dear Reviewer,

Thank you for taking the time to review our revised manuscript reviewing our manuscript.  Your insightful comments, along with those of other reviewers, were consistently helpful in improving the quality of our paper.

We are pleased to inform you that we have carefully considered each of your comments and have made the necessary revisions to the manuscript. We have addressed your concerns in a point-by-point response.

We greatly appreciate your effort in offering invaluable suggestions to review our manuscript. Your contributions have undoubtedly made our work more clear highlighting the added value, and we are very grateful for your support.

Reviewer Comments:

Reviewer 2

This is a research about covid and AKI. Much have been published already in terms of AKI covid. The positives of the study include using machine learning for risk prediction, which can be adopted in the future.

Answer: We greatly appreciate your time and effort in reviewing our manuscript and providing invaluable feedback. Your contributions have undoubtedly made our work more clear, and we are grateful for your support.

#1. Add literature about COVAN, ApoL1, collapsing GN and Covid

A: We thank the distinguished reviewer for highlighting this aspect. The following data was included into the manuscript, please note the lines 438-463 “The COVID-19 pandemic has also revealed reports of collapsing glomerulopathy similar to those documented during the HIV epidemic. In terms of the clinical picture, these patients present AKI and nephrotic-range proteinuria. The histopathological examinations of the lesions were similar to those seen in other forms of collapsing glomerulopathy (segmental/global collapse of the glomerular capillary tuft, hypertrophy in association hyperplasia of the podocytes and parietal epithelial cells, and protein resorption droplets within hyperplastic glomerular epithelial cells). This type of nephropathy should be distinguished from most cases of AKI in SARS-CoV-2 infection, which are characterized by acute tubular injury. Tubulointerstitial lesions typically associated with apolipiprotein L1 (APOL1)-related disease also include microcystic tubular dilatation and tubular injury. COVID-19-associated nephropathy (COVAN), may impact individuals in some regions of the world, this entity should be particularly suspected in cases of African descent patients who present with SARS-CoV-2 infection, AKI and nephrotic-range proteinuria. Individuals of African ancestry present an increased risk of chronic kidney disease (CKD) and kidney failure in the context to the presence of a polymorphisms in the APOL1 gene.” Ref. 34-36

#2. Menez at al have included significant biomarkers differences in Covid aki vs regular AKI.

A: We thank the distinguished reviewer for suggesting this reference. The following data has been included in the manuscript (lines 491-516) “Patients hospitalized with SARS-CoV-2 infections are at increased risk for developing long-term adverse health outcomes, part of this patients will present long-term kidney dysfunction. Identifying this category of patients that are at high risk for adverse kidney events is important in terms of a nephrology follow-up. From a prospective cohort of 576 patients hospitalized with COVID-19, Menez et al., report that the plasma biomarkers soluble tumor necrosis factor receptor 1 (sTNFR1) and sTNFR2 measured in admitted patients with COVID-19 were associated with a greater risk of adverse kidney outcomes, being strong predictors for this events. Also patients who are hospitalized with COVID-19 and have increased urinary biomarkers like neutrophil gelatinase-associated lipocalin, monocyte chemoattractant protein, and kidney injury molecule 1 are at high risk for adverse kidney outcomes and death.” Ref 37-38

#3. Line 25 in abstract not clear

A: Thank you for pointing out this fact. We performed the necessary changes so that the statement is clearer now. Please note line 25: “We noted statistically significant differences between the two study groups related to different tissues lesions(LDH), particularly at the pulmonary(CT-severity score), hepatic(AST, ALT), and muscular levels(Creatine-kinase).”

#4. Would describe more about CART in the introduction and abstract. Otherwise, most of the study is already well known.

A:  We thank the distinguished reviewer for the suggestion.

We succinctly added some CART details in abstract and introduction and materials and methods.

Please note in the abstract the lines 21-23 ….” The latter was constructed in a reverse manner, starting from the top with the root and branching out until the splitting ceases, interconnecting all the predictors to predict the overall outcome(AKI vs non_AKI).” and the lines 30-32 “ CART algorithm approach yielded decision paths suggesting three categories at risk for AKI progression: low risk (0-40%), medium risk (40-80%), and high-risk category (>80%).”

Please also to refer to the added lines(207-224) in the materials and methods : “ The CART algorithm functions as a dichotomous decision tree that employs the Gini index and entropy principle to divide data according to predictive variables and node purity, advancing from parent to offspring nodes. The optimal solution, which substantially enhances node purity, is selected from a range of plausible partitioning paths. This iterative process continues until either the stipulated cessation criteria are fulfilled or any further decrease in node impurity becomes unattainable. The central objective revolves around pinpointing the supreme partition point (threshold value) for a predictive variable, amplifying the partitioning criteria in line with the Gini index, Twoing measure for impurity with categorical variables, or the LSD (Least Squares Deviation) measure for impurity in the context of continuous variables. The algorithm then ascertains the most fitting node division by opting for the predictive factor that maximizes the partitioning criterion, culminating in the most notable reduction in node impurity. This procedure is reiterated for every child node, ceasing only when further enhancements are unattainable or predetermined halting criteria are satisfied. Typically, the minimum threshold for advancement is determined by the user and is often positioned at 0.0001. The CART decision tree demonstrates adaptability in accommodating diverse data types and distributions, resilience against anomalies, and adept handling of missing data by means of alternate divisions through its fully automated mechanism.”

#5. Line 40- not clear

A: Thank you for pointing out this fact. We performed the necessary changes so that the statement is clearer now. Note the lines 54-56

“ Around 20-50% of the patients monitored in the intensive care unit (ICU) generally develop some form of renal impairment during their hospitalization.”

#6. Line 43 Acute renal impairment- inconsistent use of definition

A: Thank you for the this suggestion regarding the consistency of acute kindey injury . We performed the necessary changes referring to acute kidney injury with regards to prevalence. Please notice the new lines 57-59

The prevalence of acute kidney injury associated with COVID-19 was estimated to be between 0.5 and 85% [4], with a mortality rate of approximately 93% [5].”

#7. Is there any information about baseline proteinuria and also data on albuminuria

A: We appreciate this question and we agree that these variables would have strengthen even more our findings, however we did not include any information on proteinuria/albuminuria due to large missing data in this respect. It was not regularly dosed, unfortunately.

#8. One of the most common manifestations of covid and kidneys is hematuria, nothing is mentioned in terms of long term prognosis. 4 years into pandemic, i think it would provide more useful information

A: We highly agree with the distinguished reviewer’s opinion. Progression to chronic kidney disease and long-term prognosis would have been indeed useful information as noted. Nevertheless, we didn’t find in our records any information regarding the percentage of discharged patients who experienced progression to chronic kidney disease or data regarding the long-term prognosis including hematuria. Therefore, we mentioned this point as a study limitation that restricted our ability to assess the long-term outcomes and determine the extent of chronic kidney disease development in the studied population. Studying long-term prognosis might be a subject for a future study. From our clinical experience we believe it is very important to acknowledge that hematuria was also observed in non-AKI cases, particularly when the patient had fever. The differentiation of these cases without a quantitative evaluation via the Addis method presented its own set of difficulties and we didn’t find systematic recordings in this respect.

#9. would provide more information about Sibiv Hospital, how big, catchment area, referral centers, how many bed ICU and inpatient services

A: Please note that we added the suggested information in the lines 89-96 : The County Clinical Emergency Hospital Sibiu, Romania, is a multidisciplinary academic hospital with 1054 beds. It has been selected on the list of national hospitals serving for the treatment of COVID-19 patients from the start of the COVID-19 pandemic. On normal bases the hospital also admits complex cases from the surrounding counties. The maximum capacity for the management of COVID-19 patients was around 100 beds and 28 ICU beds were dedicated to COVID-19 patients, beside the nonCOVID-19 ICU beds (24) and dedicated beds in some other departments (cardiology, neurology, obstetrics).

#10. The eligibility criteria is not clear

A: Thank you for raising this issue. We value your input and have carefully examined the inclusion/exclusion criteria and the flowchart against our specific objectives. Based on our assessment, we have concluded that they were properly aligned with our goals however we are open to any further suggestion to bring more clarity in this respect. Kindly note that we are in line with other author reports.  Please refer to the lines 118-129:

“ Eligible patients met cumulatively the following criteria: i) Adult patients aged over 18 years and ii) patients diagnosed with SARS-CoV-2 infection via Real-time reverse transcription-polymerase chain reaction (RT-PCR) positive for SARS-CoV-2, and iii) patients with AKI during the study period (April 2020 and November 2021). Exclusion criteria: i) Patients with chronic kidney disease and/or ii) patients known with previous renal replacement therapy(prior to admittance) and/or iii) patients whose records lacked in proinflammatory markers or and/or iv) patients with asymptomatic forms of COVID-19, and/or v) all the other cases that couldn’t be balanced on age and gender with the AKI group. Inclusion/exclusion criteria are summarized in figure 1.

We discarded CKD patients from our study as CKD is defined by markers of kidney damage or decreased glomerular filtration rate (GFR) persisting for >3 months prior to admittance and is classified according to cause, GFR, and albuminuria criteria (CGA classification) [11,12].

#11. What is the importance of creatinine ratio

A: Thank you for highlighting this point.  According to KDIGO guidelines we have referred to the creatinine ratio as the peak creatinine value relative to the baseline value[Ref-29]. We have properly addressed the typo in the lines 304-305 “ creatinine ratio (the maximum value urea and creatinine values relative to the baseline).

# 12. Lines 90-95 not clear

A: We thank the distinguished reviewer for highlighting this aspect. We hope that we have successfully addressed this issue in our response addressed to comment #10. Please note the the new lines 118-126.

We hope that the revised form of the manuscript and our accompanying responses properly addressed all the raised issues making our manuscript suitable and ready for publication in Healthcare. We shall look forward to hearing from you at your earliest convenience.

With our best regards,

Sincerely yours,

Victoria Birlutiu, Prof. Habil. M.D. Ph.D

Bogdan Neamtu, Associate Professor, M.D. Ph.D.

Rares Mircea Birlutiu, M.D. Ph.D.

Reviewer 3 Report (New Reviewer)

I am satisfied with the design and content of the project. This concerns population of adults with COVID-19 infection and provides clear data about the risk of complications in the course of the disease.

The clinical background is clear and justified. Methods adequately chosen and described. 

Results well presented and sufficiently described. Discussion well led, however to much impact given to the hypothesis of direct SARS virus attack on kidney tissue when no biopsy or autopsy data had been presented. 

Conclusions well described.

Abstract should be more precisely written to present detailed conclusions.

Author Response

Dear Reviewer,

Thank you for taking the time to review our revised manuscript reviewing our manuscript.  Your insightful comments, along with those of other reviewers, were consistently helpful in improving the quality of our paper.

We are pleased to inform you that we have carefully considered each of your comments and have made the necessary revisions to the manuscript.

We greatly appreciate your effort in offering high-quality suggestions to review our manuscript. Your contributions have undoubtedly made our work more clear highlighting the added value, and we are very grateful for your support.

Reviewer Comments:

Reviewer 3

I am satisfied with the design and content of the project. This concerns population of adults with COVID-19 infection and provides clear data about the risk of complications in the course of the disease.

The clinical background is clear and justified. Methods adequately chosen and described.

Results well presented and sufficiently described. Discussion well led, however to much impact given to the hypothesis of direct SARS virus attack on kidney tissue when no biopsy or autopsy data had been presented.

Conclusions well described.

Abstract should be more precisely written to present detailed conclusions.

Answer: We greatly appreciate the distinguished reviewer’s time and effort in reviewing our manuscript and providing invaluable feedback. We highly agree with the fact that much impact given to the hypothesis of direct SARS virus attack on kidney tissue when no biopsy or autopsy data had been presented. However we strived to offer possible hypotheses for the pathological mechanisms that accounted for the biomarkers related to acute kidney injury during Covid-19 inflammatory response. Please also note we included more details in the abstract to present the conclusions of our study.

We hope that the revised form of the manuscript and our accompanying responses properly addressed all the raised issues making our manuscript suitable and ready for publication in Healthcare. We shall look forward to hearing from you at your earliest convenience.

With our best regards,

Sincerely yours,

Victoria Birlutiu, Prof. Habil. M.D. Ph.D

Bogdan Neamtu, Associate Professor, M.D. Ph.D.

Rares Mircea Birlutiu, M.D. Ph.D.

Round 2

Reviewer 2 Report (New Reviewer)

Appreciate the changes. 

This manuscript is a resubmission of an earlier submission. The following is a list of the peer review reports and author responses from that submission.

Round 1

Reviewer 1 Report

The main limitation of the study is not only in retrospective analyzing the data in single hospital, but primarly of presenting the numerical and continuous variables (K-levels?) with their statistics of for example laboratory analysis. The most problematic are laboratory parameters in Table 4. (IL-6 , NLR, procalcitonin etc.) where abbreviation N is unknown and SDs are huge or missing (IL-6). Upon these numbers and statistics is difficult to calculate significant difference between control and study group and analyze the outcomes using univariate analysis and CART decision-tree approach. Such machine learning approach is acurate only with the reliable input data.

Reviewer 2 Report

In this manuscript, the authors summarized and discussed 466 cases of SARS-CoV-2 infection with or without AKI, and give a brief picture of the association of particular risk factors with AKI progressing in COVID-19 patients. The manuscript is well-organized and clearly stated. I would suggest accepting it after the following minor concerns are addressed.

1. The research utilized CART decision-tree approach for data analysis, while limited information about it is showed in the background. Why do you choose to use this approach and what is its strength compared with other methods?

2. Is there any imaging or pathological information of the patients in the study that support your conclusion?

3. There are several format mistakes in the manuscript, including missing space between number and unit in line 91, extra text-indent in line 334, etc. Please review the whole manuscript thoroughly to avoid the same mistakes.

The manuscript is well-organized and clearly stated. The quality of English language is good while several format mistakes should be avoided as I mentioned above.

Reviewer 3 Report

Thank  you  for the opportunity  to  review  this manuscript.

It is a very interesting and  valuable  paper with some improvements  to  be  made:

1)Overall, English   grammar  may  be  improved, so  revision from  either  a native  speaker  or  professional  translator  should  be  made.

2)In the  introduction, “Renal impairment“ should  be  changed  to a  term  with a standardized operational   definition.

3)In the  Materials  and Methods  section:

Add a  figure  showing  the  patients  selection process.

Briefly describe  if  you discarded  also patients   with  any CKD.

Specify the  method  used   to asses  normal distribution of  data.

Specify the statistical probe  used  to  asses  association  among   categorical  variables

Specify the operational  definition of AKI  used

4) In the  results  section:

Specify the  frequencies of  renal  outcome in terms  of  renal  recovery, progression  to  acute  kidney  disease,  progression to  chronic  kidney disease,  renal replacement  therapy  need

Specify  how  numerical  variables  were used  to  describe gender (instead  of  dichotomic  variables) and  intensive care  admission.

4)In the  statistical analysis   section:

Specify  the  central  tendency  measurements  used  according t o type  of  variables in order  to assure  the  pertinent  statistical test  was  used.

5) In the conclusion section: Review  that  conclus.ion  must  be  concordant  with  the  result  reported.

Review  by a native  speaker of professional  translator needed.